# Identification of Azoxystrobin Glutathione Conjugate Metabolites in Maize Roots by LC-MS

**DOI:** 10.3390/molecules24132473

**Published:** 2019-07-05

**Authors:** Giuseppe Dionisio, Maheswor Gautam, Inge Sindbjerg Fomsgaard

**Affiliations:** 1Department of Molecular Biology and Genetics, Research Center Flakkebjerg, Aarhus University, Forsøgsvej 1, 4200 Slagelse, Denmark; 2Department of Agroecology, Research Center Flakkebjerg, Aarhus University, Forsøgsvej 1, 4200 Slagelse, Denmark; maheswor.gautam@gmail.com (M.G.); inge.fomsgaard@agro.au.dk (I.S.F.)

**Keywords:** azoxystrobin, glutathione, glutathione conjugate, untargeted metabolomics

## Abstract

Xenobiotic detoxification in plant as well as in animals has mostly been studied in relationship to the deactivation of the toxic residues of the compound that, surely for azoxystrobin, is represented by its β-methoxyacrylate portion. In maize roots treated for 96 h with azoxystrobin, the fungicide accumulated over time and detoxification compounds or conjugates appeared timewise. The main detoxified compound was the methyl ester hydrolysis product (azoxystrobin free acid, 390.14 *m*/*z*) thought to be inactive followed by the glutathione conjugated compounds identified as glutathione conjugate (711.21 *m*/*z*) and its derivative lacking the glycine residue from the GSH (654.19 *m*/*z*). The glycosylated form of azoxystrobin was also found (552.19 *m*/*z*) in a minor amount. The identification of these analytes was done by differential untargeted metabolomics analysis using Progenesis QI for label free spectral counting quantification and MS/MS confirmation of the compounds was carried out by either Data Independent Acquisition (DIA) and Data Dependent Acquisition (DDA) using high resolution LC-MS methods. Neutral loss scanning and comparison with MS/MS spectra of azoxystrobin by DDA and MSe confirmed the structures of these new azoxystrobin GSH conjugates.

## 1. Introduction

Azoxystrobin is one of the widely used fungicides in plants. The metabolism of azoxystrobin in plants is well described [1] and recently our laboratory has reported mass spectrometric methods to identify and quantify azoxystrobin metabolites in plants [2,3,4]. However, glutathione conjugation of azoxystrobin in plants has not been reported yet [1,5]. The only known report on GSH conjugation of azoxystrobin comes from an animal metabolism study where azoxystrobin was reported to conjugate with glutathione in the cyanophenyl aromatic ring with the electrophilic carbon in the –ortho position to the cyano group [6]. However, a later in vitro experiment undertaken to conjugate azoxystrobin with glutathione by plant-derived glutathione transferase was unsuccessful [5]. Glutathione (GSH) conjugation is a crucial step in the detoxification process of xenobiotics in living organisms [5,6,7]. GSH conjugation occurs in the cytoplasm of plant cells when electrophilic centers in xenobiotics are attacked by the nucleophilic thiol (–SH) group of glutathione [5]. This process increases the solubility of xenobiotics, which can thus be transported from cytoplasm to vacuoles. As increasing concentration of GSH-xenobiotic could be potentially harmful for plant cells, the conjugates are sequentially degraded in vacuoles by enzymatic actions thereby enabling plants to tackle harmful effects of xenobiotics [7]. During the degradation of GSH-xenobiotic conjugate in vacuoles, glycine residue from GSH is cleaved off enzymatically, which results in glutamic acid and cysteine residue of GSH attached to the xenobiotic [5,7].

GSH detoxification has been shown to occur in xenobiotics compound by conjugation, at a structural level, to aliphatic, aromatic, benzylic, disulfide, and thioester groups [8]. However, no other conjugation than aromatic conjugation of azoxystrobin with GSH has been reported or studied so far. We hypothesized that the carbon in the –CN group of azoxystrobin is a favorable candidate for the nucleophilic attack by the –SH group in GSH. The synthesis of analytical standards of such conjugates of GSH with azoxystrobin was out of the scope of the present study. Thus, we aimed to verify the presence of GSH-azoxystrobin conjugates at the –CN group by suspect screening of such metabolites and others possible ones by using high-resolution mass spectrometry. Our initial candidate structure for GSH conjugates were aromatic and cyano conjugation with azoxystrobin. However, the algorithm used here to detect GSH conjugates of azoxystrobin, a priori of its structure determination, is to detect at MS/MS level the simultaneous presence of GSH fragments neutral loss and the main azoxystrobin MS/MS fragments. Untargeted LC-MS/MS analysis was automated by using Progenesis QI for metabolomics where the candidate MS scans were compared also over time for increased appearance due to the plant detoxification of azoxystrobin. Once we had identified by MS/MS spectral matching we did confirm its reproducibility in detection by setting up a method to use in future for targeted detection by quadrupole-linear ion trap mass spectrometry to develop sensitive triple-quadrupole mass spectrometry-based methods for quantification of GSH-azoxystrobin conjugates.

The conjugation of GSH to xenobiotics is operated enzymatically by plant Glutathione Transferases (GSTs), which is a multi-functional superfamily of detoxification enzymes [9]. They possess a hydrophobic binding domain (H-site) and the ability to bind glutathione (G-site). The greatest similarity between members of the GST family is observed in the GSH binding domain containing a motif of four highly conserved amino acids [10]. Here, a catalytically essential tyrosine or serine activates glutathione by lowering the pKa of the thiol group from around pH 9 to approximately pH 6, thus enhancing the rate of nucleophilic attack by the resulting thiolate anion toward electrophilic substrates at physiological pH [11,12,13]. The H site is more variable, accounting for the broad range of endogenous and xenobiotic substrates utilized by this family of GSH dependent enzymes [13].

Despite the fact that reduced glutathione (GSH) is capable of undergoing spontaneous conjugation with compounds containing electrophilic centers for the detoxification of several xenobiotics compounds, different classes of GSTs are, in fact, involved because they can lower by several fold the Km related to the nucleophilic thiolate anion attack of GSH to the electrophilic substrates, with concomitant displacement of a leaving group (i.e., a proton) which more rarely takes part in addition reactions [14].

In our former study, we established a maize root culture model system to study the biotransformation of azoxystrobin in plants [3]. In the current study, we report on more results from the use of maize root culture as a model system to study not only the demethylated metabolite, azoxystrobin free acid (AzFA), which represents the main plant metabolite but also, in lower extent of abundancy, the GSH-azoxystrobin conjugates.

## 2. Results

The first observation to note in treated maize roots samples as compared to control is the accumulation of azoxystrobin (monoisotopic mass 403.1168), as expected, judged by looking at the appearance of the peak at 404.1197 *m*/*z* representing MH+ of azoxystrobin in the MS survey profile (Figure 1), that is accumulating after 96h as compared to the control. Further evidence that the peak of 404.1197 *m*/*z* eluting at about 28.5 min represents for real the MH+ of azoxystrobin is given by its fragmentation pattern when observing its MS/MS profile (Figure 2). Furthermore, the major methyl ester hydrolysis product of azoxystrobin, azoxystrobin free acid, was differentially present in the treated samples as compared to the controls (Appendix A).

The conjugated azoxystrobin metabolite analysis relied on the fact that the MS/MS spectra of the main peaks of azoxystrobin-conjugates should contain, at least, one of the most abundant MS/MS peak as shown in Figure 2. Using Masslynx function for extracting selected peaks from a chromatogram, it is possible to create one extracted chromatogram profile at survey as well as at MS/MS level. By scanning the MS/MS fragment in order to filter out peaks with 372 *m*/*z* from the MS/MS spectra of the treated versus untreated controls it was possible to visualize the presence of four main peaks at MS/MS level that contains the species 372 *m*/*z* (Figure 3). Four potential azoxystrobin metabolites containing at MS/MS level the 372 *m*/*z* species were found at retention time (RT) respectively: (a) 18.36 min; (b) 18.68 min; (c) 19.72 min; and (d) 30.5 min, while RT 28.79 min was related to the azoxystrobin itself.

Matches were found in relation to the metabolites proposed to be azoxystrobin-glutathione conjugates at the cyano group due to the bonding of sulphur with the electrophilic carbon with monoisotopic mass 710.200 (MH+ 711.30 *m*/*z*) and after glycil neutral loss 653.1792 (MH+ 654.19 *m*/*z*) (Figure 4). The peak eluting at retention time (RT) 18.36 min is therefore compatible with precursor ion 711.30 *m*/*z* (Figure 5) and the peak eluting at RT 18.67 min could be represented by the same azoxystrobin-glutathione conjugate with glycil neutral loss, represented by the precursor ion 654.19 *m*/*z* (Appendix A). The correct identification of potential azoxystrobin conjugate at RT 19.72 min was indicated by the neutral loss of 162 Da (glucosyl) and corresponds to azoxystrobin free acid glucoside with a monoisotopic mass of 551.1540 Da (MH+ 552.19 *m*/*z*) (Appendix A). An analysis with QTRAP IDA method revealed similarity in MS/MS spectra for 711.3 *m*/*z* including the evidence for a neutral loss of 129 corresponding to the loss of glutamic acid residue as shown in Appendix A. Finally, the correct identification of potential azoxystrobin conjugate at RT 30.55 min was revealed by the neutral loss discovery of demethylated azoxystrobin-GSH conjugate of the precursor specie 697.48 *m*/*z* (Appendix A).

The presence of these metabolites of azoxystrobin in maize roots was also investigated by differential untargeted metabolomics using Progenesis QI ver. 2.0. A database of SDF structures to search such metabolites was custom made from known and possible metabolites and validation of such metabolites was performed by the internal Metascope theoretical fragmentation engine using DDA MS/MS fragmentation profile both at MS1 and the most abundant MS/MS fragments.

The top relative maximum abundance is shown in Table 1 and it was recorded respectively for compounds at *m*/*z* 404.1068, 372.0798, 549.1369, 549.1715, 654.1935, 711.2189, 557.1727, and 552.1740. The validation of each peak was performed by looking at the control versus the treated root MS/MS profile for each *m*/*z* species above which a signature of azoxystrobin MS/MS spectra could be found.

Not all the metabolites of azoxystrobin were found by the Progenesis Metascope engine but at least the main azoxystrobin conjugates were detected in order of significance (ANOVA) and abundance. The peaks were presented by precursor ions 654.1935 *m*/*z*, 711.2189 *m*/*z*, and 552.1740 *m*/*z*, corresponding respectively to azoxystrobin-GSH conjugate with glycil loss, azoxystrobin-GSH conjugate, and azoxystrobin free acid glucoside conjugate (Figure 6). Time course accumulation of selected metabolites have been chosen for visualization in Progenesis QI where the untreated control (CTRL) and the azoxystrobin treated samples (AZO) have been analyzed at 3 h, 24 h, 48 h, and 96 h (Figure 6). The typical representation of the trend of *m*/*z* over time and its isotope distribution can be visualized in 2D or 3D and the related area quantified by relative units which are normalized by abundance related to a standard (i.e., CTRL) in Figure 6.

All the rest of Progenesis identified peaks at MS1 containing at MS/MS level the 372 *m*/*z* species, shown in Table 1, were not validated because they were also present in the control roots extract and lacked full similarity to all MS/MS spectra of azoxystrobin.

The QTRAP MS was hence used to setup a quantification method to be further implemented in future screening of the 711.3 *m*/*z* metabolite in plant once it would be commercially available to be used as standards. The MS conditions used for the future QQQ detection of 711.3 *m*/*z* metabolite are summarized in Table 2.

## 3. Discussion

Azoxystrobin is a strobilurin fungicide that was developed by Zeneca UK Ltd., in 1992. Its mechanism of action in fungi and bacteria rely on inhibiting the electron transport system by binding the Qo site of mitochondrial cytochrome bcl complex to inhibit microbial respiration [15]. The toxophore (chemical group that produces the toxic effect) portion of the molecule is actually not the cyano group, but the β-methoxyacrylate portion that is double methylated and that, hence, should represent the immediate sensitive region of the molecules for enzymatic deactivation by conjugation or demethylation. One of the azoxystrobin environmental degradation products, azoxystrobin free acid, was present in the methanol extracts of maize roots for the treated samples as compared to the controls (Appendix A) indicating an active demethylation mechanism of detoxification occurring *in vivo*. This initial demethylation opens the molecule to a definitive inactivation of its toxophore by glycosylation as was shown by the presence of the peak with 552.1740 *m*/*z* (Appendix A).

The conjugation of azoxystrobin with glutathione by enzymatic action of glutathione S-transferase enzyme has not been described yet in plants, although a glutathione (GSH) conjugate of azoxystrobin has been identified in rats where sulphur is conjugated to the phenolic ring in the ortho position relative to the cyano group where the conjugate species have shown *m*/*z* values of 708.18 and 694.16 respectively for the fully methylated and demethylated GSH azoxystrobin conjugate [6] not found here in our experiments. Here, we present mass spectrometric evidences that GSH was conjugated with azoxystrobin through the –CN group thereby potentially allowing detoxification of azoxystrobin. We have also proposed chemical structures of the corresponding precursors of *m*/*z* 711.2189 (Figure 4 and Figure 5) and 654.1935 (Appendix A) as being full azoxystrobin-GSH and azoxystrobin-cysteine-γglutamic acid conjugates. During the MS survey at low energy, the glycine neutral loss is already present in the MS/MS spectra of *m*/*z* 711.2189 (Figure 4), but the elution of the peak with *m*/*z* 654.1935 is at different time than that with *m*/*z* 711.2189 meaning that the loss could be happening *in vivo*. In barley, Glutathione S-conjugates accumulates in the plant vacuole and that the first step of its degradation, the formation of the respective T-glutamylcysteinyl-S-conjugate, is catalyzed by a vacuolar carboxypeptidase [16]. Following the action of a vacuolar carboxypeptidase that degrades GS-conjugates by cleaving the glycine, it is also reported a removal of the amino terminal glutamic acid residue, cleaved also successively enzymatically [17]. Besides, a membrane associated γ-glutamyl transpeptidase (γ-GTase) has been identified [18], as part of the transport from the vacuole to the cytosol, that then removes the terminal glutamic acid leaving the S-cysteinyl derivative [7]. The loss of the glycine residue from glutathione conjugates of a xenobiotic is well documented [5,19] and is known to occur during transport of xenobiotic glutathione conjugates from cytosol to vacuoles (for review see [5]). Alongside the glycine neutral loss (Δ57) (Figure 4), γ-Glutamic acid neutral loss (Δ129) has been detected (Appendix A) confirming the nature of the analyte with precursor ion *m*/*z* 711.2189. The neutral loss of 129 Da corresponding to the loss of glutamic acid from GSH conjugates is previously described [20] and validated by the method for simultaneous screening of GSH and CN adducts using precursor ion (PI) and neutral loss (NL) scans dependent product ion spectral acquisition and datamining tools on QTRAP mass spectrometry [21]. Here, hence, we have demonstrated that the 711.3 *m*/*z* metabolite possesses a 129 Da neutral loss (Appendix A) and that can be quantified by QQQ MS methods (Table 2) once the standard is available as pure compound.

Naturally occurring isothiocyanate compounds present in cruciferous vegetables, e.g., sulforaphane, are powerful inducers of glutathione S-transferase activities in laboratory animals, meaning that the GSH conjugation at the cyano group is operated enzymatically and the reactive group is represented by the carbonyl portion of the isothiocyanate [22]. Similar results have been shown for phenyl isocyanate detoxification by glutathione conjugation [23]. Here we have proposed and have shown evidence that glutathione is conjugated to the electrophilic carbon of the cyano group (–CN) of azoxystrobin. Further experiments are currently underway in order to identify possible enzymes, i.e., gluthatiose S-transferases, that might mediate in vivo GSH conjugation and detoxification of azoxystrobin in maize roots.

## 4. Materials and Methods

### 4.1. Chemicals

Analytical grade azoxystrobin (purity 99.4%) and sucrose (≥99.5%) were purchased from Sigma-Aldrich (Copenhagen, Denmark). Azoxystrobin free acid ((*E*)-2-(2-((6-(2-cyanophenoxy)pyrimidin-4-yl)oxy)phenyl)-3-methoxyacrylic acid) (10 ng/µL in acetonitrile) was procured from Dr. Ehrenstorfer GmbH (Augsburg, Germany). Acetonitrile and methanol (TOF/MS grade) were obtained from Fisher Chemical (Roskilde, Denmark). Milli-Q water was generated from a Milli-Q system (Millipore, MA, USA). Murashige and Skoog (M&S) media was purchased from Duchefa Biochemie (Harlem, The Netherlands).

### 4.2. Maize Root Culture Growth

Maize seeds (*cv*. Kaspian) were purchased from KWS Scandinavia (Vejle, Denmark) and maize roots were cultured as explained in our earlier study [3]. Briefly, maize seeds were surface sterilized with ethanol and sodium hypochlorite and rinsed with sterilized Milli-Q water. The seeds were transferred into petri plates containing sucrose and full-strength M&S media that had been solidified with phytagel. The pH of the media was maintained at 5.8. The plates were incubated under light/dark condition for 16/8 h at 23 °C for seven days. The roots from germinated seeds were detached and transferred into Erlenmeyer flasks containing half-strength M&S and sucrose where the pH was maintained at 5.8. The flasks were covered with aluminum foil to maintain a dark condition within the flasks. The flasks were shaken at 100 rpm at 25 °C for seven days and the maize root culture was ready for azoxystrobin uptake and metabolism study.

### 4.3. Azoxystrobin Uptake

Azoxystrobin was dissolved in methanol and added to sterile half-strength liquid M&S growth medium (pH 5.8) contained in a beaker to yield a final azoxystrobin concentration 100 µM in 500 mL medium as explained in Gautam et al. (2018). Maize roots were transferred into the beaker and swirled gently to mix azoxystrobin thoroughly. A beaker with similar content except azoxystrobin was used as control. Azoxystrobin treatment continued for three hours after which a portion of roots (5–10 g) was cut-off with sterile scissors, washed thoroughly with sterile water, wiped with tissue paper, wrapped in aluminum foil, and flash frozen in liquid nitrogen. Control maize roots was sampled similarly after three hours. The samples were labelled as 3 h sample.

The treatment solution was decanted off, maize roots were first washed with sterilized Milli-Q water and then with M&S medium, and finally transferred to sterile liquid M&S medium devoid of azoxystrobin. The control was also washed and transferred similarly. The flasks were wrapped with aluminum foil to prevent exposure to light and allowed to stand in a sterile bench throughout the uptake experiment period. Samples were collected after 3, 24, 48, 72, and 96 h of initial exposure with azoxystrobin.

### 4.4. Sample Preparation and Extraction

Frozen maize roots were freeze dried on a DryWinner 6-85 freeze drier (Holm & Halby, Brøndby, Denmark) while still wrapped in aluminum foil. The freeze-dried roots were ground in GenoGrinder 100 (Spex, Metuchen, NJ, USA). 20.0 mg of homogenized root tissue was weighed in 2-mL Eppendorf tube and suspended in 1 mL of 50% methanol (*v/v*) containing 0.1% formic acid. The tube was shaken vigorously, ultrasonicated for 15 min, and centrifuged at 16,000× *g* at 20 °C for 15 min in a Sigma 1-14K microcentrifuge (Buch and Holm, Herlev, Denmark). The supernatant was filtered with 0.22 µm KX Syringe PTFE filter (Mikrolab, Aarhus, Denmark) and analyzed immediately with or stored at −20 °C before analysis.

### 4.5. LC-MS/MS Analysis

#### 4.5.1. Q-TOF Analysis

Methanol content in the extracts were adjusted to 25% before injecting. The liquid chromatography system was composed of a nano Acquity UPLC (Waters, Milford, CT, USA) equipped with a nano Trap column (Waters XSelect™ HHS T3, 5 µm beads diameter, 180 µm × 20 mm) for accumulating the injected samples and an analytical capillary column with the same chemistry (Waters nano RP-C18 HHS T3 type, 1.8µm, 150µm × 100 mm) for sample separation. The nano capillary column was attached with a nano ESI source to a Waters Q-TOF premiere (Waters, Milford, USA). The Q-TOF operational conditions were the following: 3.2 kV for capillary cone voltage and extraction, 30 kV as sampling cone voltage, and 3.4 kV as ion guide voltage, and the source temperature was set to 120 °C. The instrument was calibrated in V positive mode using the MS/MS profile of M2H + 785.8426 *m*/*z* GluFib-B ([Glu1]-Fibrinopeptide B human, F3261, Sigma Aldrich, Merck, Germany) as Z-lock mass (reference calibration) at 23 eV as collision energy obtaining a calibration range from 72 to 1286 Da, and with a ppm error of +/− 2.

The entire length of the LC run was 42 min. Mobile phase A was 0.1% formic acid (FA, Sigma Aldrich, Germany) and phase B was acetonitrile (ACN, Thermo-Fischer Scientific, Hvidovre, Denmark) with 0.1% FA. The gradient conditions were from 15% phase B to 70% phase B in 25 min and from 25 min to 27 min from 70% phase B to 85% phase B; from 27 to 30 min, the percentage of mobile phase B raised from 85% to 95%. After 30 min the percentage of phase B was constant set to 95%. Data acquisition was performed in Masslynx version 4.1 (Waters, Milford, USA) either as data-independent acquisition (DIA) mode by all ion fragmentation (MSe) with MS1 and MS/MS (MS2) range from 50 to 2000 *m*/*z*, or data dependent (DDA). For the DDA mode the MS1 survey was acquired from 50 to 1000 *m*/*z* and MS/MS was acquired from 500 Da above after peak deisotoping: scan time 1.5 s with variable collision energy from 5 to 40 V.

Differential untargeted azoxystrobin metabolite, by the means of spectral counting comparison, was performed by Progenesis QI, version 2.0 (Nonlinear Dynamics, a Waters company, Newcastle upon Tyne, UK). The import of raw MSe data acquired by Masslynx was transformed into centroid data by M2H + 785.8426 *m*/*z* GluFib-B lock mass data calibration and refinement including dead time correction and charge deconvolution. Data runs, in triplicate, were hence layered into 2D graphs obtained by the retention time versus the *m*/*z* indexing of the MS1 and MS/MS peaks. After the multirun 2D alignment the peptide3D Apex3D algorithm was used to peak picking and quantification [24]. Label-free ion/adducts quantitation was performed by general all ion compound normalization using built in LOWESS (locally weighted scatterplot smoothing) algorithm and weighted spectral counting alongside the ion/adduct isotopic peaks [25]. The search algorithm was the built-in Progenesis Metascope using a custom Structure Data File (SDF) databases created by joining together many single chemical 2D structures of azoxystrobin precursors as SDF files by Progenesis SDF studio. Chemical 2D structures of azoxystrobin precursors were downloaded from ChemSpider website or manually drawn either by ChemDraw Pro (CambridgeSoft, PerkinElmer, Waltham, MA, USA) or BIOVIA Draw 2018 (BIOVIA, Accelrys, San Diego, CA, USA). The Progenesis Metascope search of the custom SDF database was carried out assuming 50 ppm error for precursor ions and 30 ppm for theoretical fragments.

#### 4.5.2. QTRAP Analysis

The LC-MS/MS analysis was done with a QTRAP 4500 mass spectrometer (AB Sciex, Framingham, MA, USA) coupled to an Agilent 1260 Infinity HPLC system (Santa Clara, CA, USA). Reversed phase HPLC was done with a BDS Hypersil C18 column (250 × 2.1; 5 µm) (Thermo Scientific, Waltham, MA, USA) using a gradient with eluent A 0.1% formic acid in Milli-Q water and B 0.1% formic acid in acetonitrile. Column compartment was maintained at 30 °C. The column was equilibrated with 20% B for 10 min in the beginning of each run. The linear gradient was increased from 20% to 95% B (0–15 min) and maintained at 95% B (15–25 min). The mobile phase flow rate was maintained at 300 µL/min with 5 µL injection volume. Samples were filtered through a 0.22 µm KX Syringe PTFE filters (Mikrolab, Aarhus, Denmark) before LC-MS/MS analysis.

The QTRAP MS was used in Information Dependent Acquisition (IDA) mode. After the suspect screening of glutathione conjugates with Q-TOF analysis, MS/MS spectra of the selected hits were acquired using IDA.

The MS/MS spectra from one to two of the most intense peaks matching the survey MRM transitions outlined in Table 1 were acquired if the peak intensity was above IDA threshold of 3000 counts per second (cps). Dynamic exclusion was applied to avoid acquiring MS/MS spectra from the same transition for the next 15 s after 3 occurrences of a transition, thus allowing for the detection of co-eluting peaks. Two EPI experiments were done for the two most intense peaks of each transition meeting the IDA criteria. EPI spectra were acquired between *m*/*z* 80 to 720 in positive ionization mode with a scan rate at 20,000 amu per second. Two EPI spectra from each transition was summed to give a final spectrum which resulted in a dwell time of 1.6 s for a complete MRM-EPI scan. During the acquisition of EPI spectra, DP was ramped between 50 to 70 V. Collision energy of 35 V and a spread of 15 V was used thus acquiring EPI at CE 20, 35, and 50 V.

## Figures and Tables

**Figure 1 molecules-24-02473-f001:**
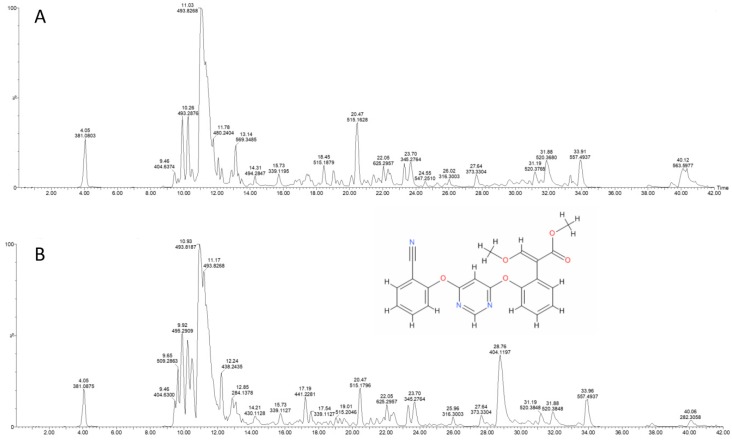
Q-TOF survey profile of the total metabolites extracted from maize root in control (panel **A**) and azoxystrobin treated ones (Panel **B**). Untargeted MS of roots methanol extracts at 96h has revealed the presence of a differential peak occurring at about 28.5 min with 404.1197 as *m*/*z* which is correspond to MH+ of azoxystrobin (exact mass, 403.1168; IUPAC Name, methyl (*E*)-2-[2-[6-(2-cyanophenoxy)pyrimidin-4-yl]oxyphenyl]-3-methoxy-prop-2-enoate). In the ordinate is shown the signal intensity in percentage; in the abscissa the retention time (RT) in minutes.

**Figure 2 molecules-24-02473-f002:**
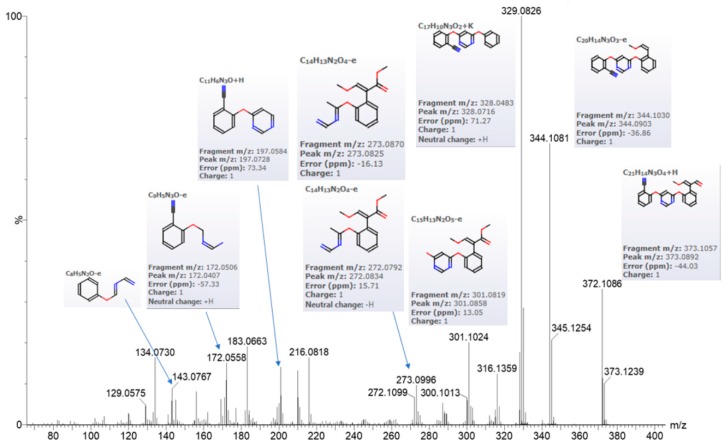
Q-TOF MS/MS DDA profile of the root metabolite extracted from maize root at 28.667 min as retention time. The precursor specie isolated by the quadrupole at 404.09 *m*/*z* was subjected to high energy ramping from 10 to 40 eV and its MS/MS profile shown here clearly indicates the presence of fragments of azoxystrobin as assessed by comparing the peak with similar fragmentation pattern from massbank (http://www.massbank.jp/RecordDisplay.jsp?id=AU324504&dsn=Athens_Univ). The peak annotation has extracted by Progenesis QI Metascope built-in fragmentation and annotation system.

**Figure 3 molecules-24-02473-f003:**
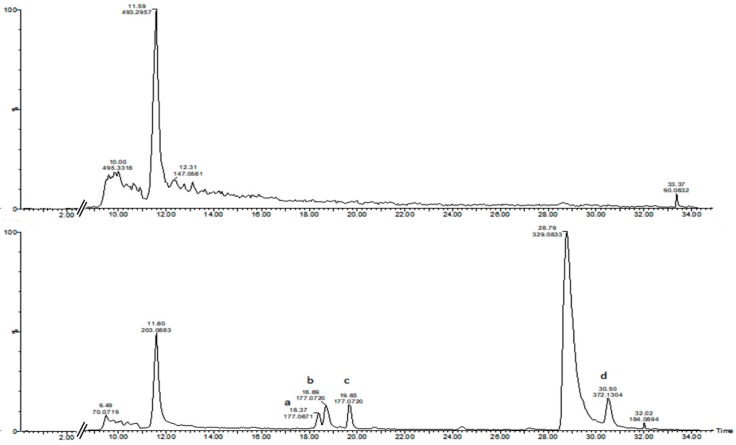
Extracted MS chromatograms at MS/MS (MSe) level of the precursor specie 372 *m*/*z*. A comparative extracted chromatogram for control untreated (**top** MS/MS trace) and treated with azoxystrobin (**lower** MS/MS trace) extracted and analyzed after 96 h. Four potential azoxystrobin metabolites are differentially displayed containing the 372 *m*/*z* species as visualized by the related extracted chromatogram at *m*/*z* 372 with different retention times: (a) RT 18.36 min; (b) RT 18.68 min; (c) RT 19.72 min; and (d) RT 30.5 min, being RT 28.79 min related to the azoxystrobin itself.

**Figure 4 molecules-24-02473-f004:**
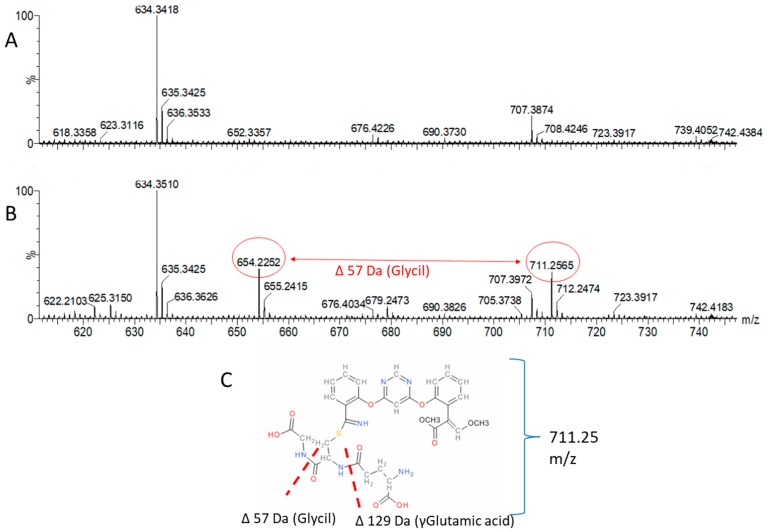
Neutral loss discovery of azoxystrobin-GSH conjugate at MS survey conditions of the precursor specie 711 *m*/*z*. Possible identification of potential azoxystrobin conjugate at RT 18.10–18.36 min. The analysis of MS survey profile for untreated (panel **A**) and treated sample (panel **B**) reveals that both 711.25 and 654.22 *m*/*z* species are present only in the treated sample. The glycil neutral loss (654.22 *m*/*z*) is already occurring at +5eV low energy level (MS1 or MS survey). The azoxystrobin-glutathione conjugate (panel **C**) at the cyano group has the MS/MS signature of azoxystrobin alongside the evidence of neutral loss of the glycil residue (neutral loss 57).

**Figure 5 molecules-24-02473-f005:**
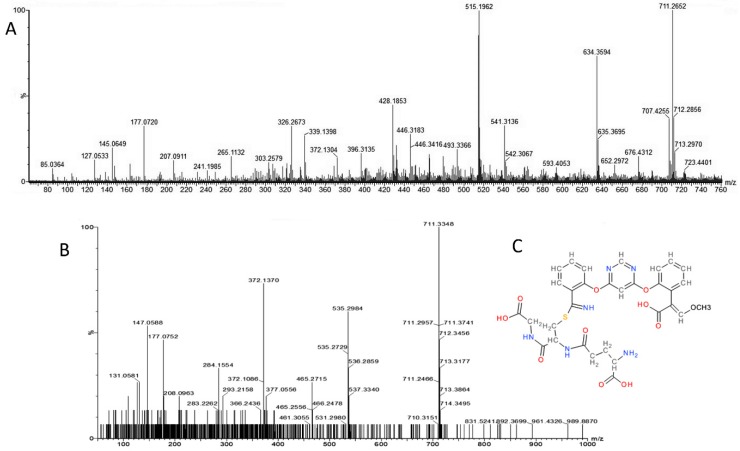
Analysis of the precursor specie 711 *m*/*z*. The identification of potential azoxystrobin conjugate at RT 18.36 min was possible by the analysis of MS survey profile (panel **A**) of azoxystrobin treated roots and its DDA MS/MS profile (panel **B**). The isolated MS/MS fragmentation profile reveals that the precursor of this azoxystrobin-conjugate/metabolite fits well with the azoxystrobin glutathione conjugate (panel **C**) at the cyano group compatible with the compound 711.30 *m*/*z* as confirmed by the presence of 372 *m*/*z* in MS/MS of the isolated peak at 711.30 *m*/*z* (panel B). Further confirmation of this compound has been done by studying the neutral losses.

**Figure 6 molecules-24-02473-f006:**
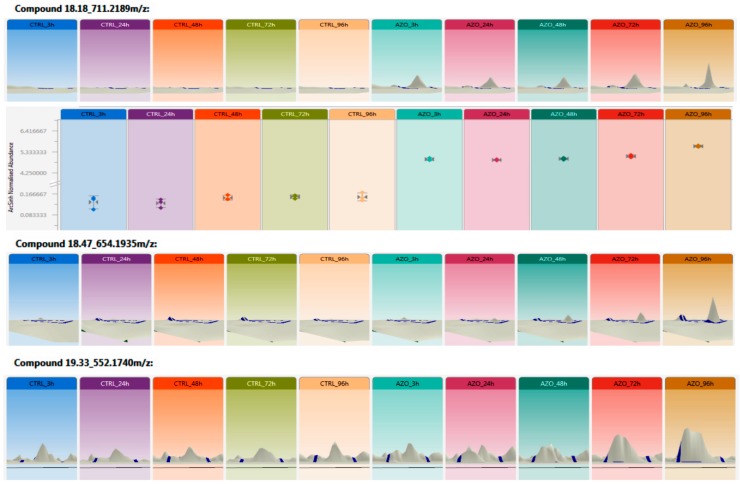
Relative Ion abundance distribution of azoxystrobin-GSH (glutathione) conjugate at MS survey conditions as quantified by the Progenesis QI software for metabolomics. The label free relative spectral counting quantification analysis (here depicted as 3D isotope abundance) has revealed the differential presence of the *m*/*z* 711.2189, accumulating over time (max at 96 h) in azoxystrobin treated roots versus the untreated control. The same trend has been detected for the peaks 654.1935 *m*/*z* and 552.1740 *m*/*z*.

**Table 1 molecules-24-02473-t001:** Possible azoxystrobin containing peaks differentially present in the treated maize root as compared untreated roots over 96 h. The searched azoxystrobin SDF structures and related metabolites present here were not all validated since most of them were also present in the control roots. Their relative max abundance as compared to the control and spectral counting is shown in the last column, where clearly the relative max abundance was recorded respectively for compounds at *m*/*z* 404.1068, 372.0798, 549.1369, 549.1715, 654.1935, 711.2189, 557.1727, and 552.1740. In bold are reported the compounds which have been confirmed being azoxystrobin metabolites.

Compound (RT_*m*/*z*)	*m*/*z*	z	RT	PW	Anova (p)	q Value	FC	HM	LM	Max Abundance
**18.18_711.2189 *m*/*z***	**711.2189**	**1**	**18.18**	**0.61**	**<1.1 × 10^−16^**	**<1.1 × 10^−16^**	**18.7**	**AZO_96h**	CTRL_24h	139.9
**19.33_552.1740 *m*/*z***	**552.1740**	**1**	**19.33**	**0.69**	**1.11 × 10^−16^**	**4.58 × 10^−15^**	**27.8**	**AZO_96h**	CTRL_48h	50.7
**18.47_654.1935 *m*/*z***	**654.1935**	**1**	**18.47**	**0.61**	**2.22 × 10^−16^**	**8.10 × 10^−15^**	**110**	**AZO_96h**	CTRL_3h	162.0
**28.70_403.0995 *m*/*z***	**404.1068**	**1**	**28.70**	**3.12**	**3.33 × 10^−16^**	**1.13 × 10^−14^**	**2.98 × 10^−3^**	**AZO_3h**	CTRL_48h	8260.1
**28.70_372.0798 *m*/*z***	**372.0798**	**1**	**28.70**	**2.94**	**5.33× 10^−16^**	**1.28 × 10^−13^**	**8.05 × 10^3^**	**AZO_3h**	CTRL_48h	6041.6
16.10_534.1777 *m*/*z*	534.1777	1	16.10	0.64	2.12 × 10^−10^	1.60 × 10^−9^	3.91	AZO_96h	CTRL_3h	57.2
23.22_557.1621 *m*/*z*	557.1621	1	23.22	1.25	4.47 × 10^−9^	2.41 × 10^−8^	6.76	CTRL_3h	AZO_96h	3340.4
13.32_743.2417 *m*/*z*	743.2417	1	13.32	0.59	5.68 × 10^−9^	3.01 × 10^−8^	3.46	AZO_96h	CTRL_3h	63.9
11.19_549.1369 *m*/*z*	549.1369	1	11.19	0.96	4.20 × 10^−6^	1.27 × 10^−5^	4.68	AZO_96h	CTRl_3h	229.8
21.12_557.1727 *m*/*z*	557.1727	1	21.12	1.17	7.97 × 10^−6^	2.30 × 10^−5^	2.39	AZO_3h	CTRl_72h	104.5
34.25_404.1264 *m*/*z*	404.1264	1	34.25	0.80	1.02 × 10^−5^	2.89 × 10^−5^	Infinity	AZO_3h	CTRL_3h	12.9
3.90_1103.2994 *m*/*z*	1103.2994	1	3.90	0.45	2.50 × 10^−5^	6.57 × 10^−5^	4.87	CTRL_3h	CTRL_48h	20.4
21.25_663.2184 *m*/*z*	663.2184	1	21.25	0.59	9.41 × 10^−5^	0.00022	2.78	AZO_3h	AZO_72h	26.5
13.21_426.1160 *m*/*z*	426.1160	1	13.21	0.85	0.000114	0.000263	1.74	CTRL_72h	CTRL_3h	94.7
13.37_549.1715 *m*/*z*	549.1715	1	13.37	2.59	0.0258	4	1.42	AZO_3h	CTRL_24h	330.3
9.32_367.0946 *m*/*z*	367.0946	2	9.32	0.51	0.0296	456	2.84	AZO_3h	CTRL_3h	57.5
1.98_404.1227 *m*/*z*	404.1227	1	1.98	0.24	0.143	0.197	Infinity	AZO_24h	CTRL_24h	14.2
3.82_370.0885 *m*/*z*	370.0885	2	3.82	0.51	0.208	0.278	2.36	CTRL_96h	AZO_96h	5.3
1.52_404.1226 *m*/*z*	404.1226	1	1.52	0.35	0.31	0.4	Infinity	CTRL_3h	CTRL_24h	3.6
9.32_355.0933 *m*/*z*	355.0933	2	9.32	0.51	0.336	0.432	4.08	AZO_3h	CTRL_3h	7.5
38.31_404.1245 *m*/*z*	404.1245	1	38.31	0.45	0.508	0.629	Infinity	AZO_48h	CTRL_3h	13.4
37.53_404.1205 *m*/*z*	404.1205	1	37.53	0.56	0.607	0.74	Infinity	AZO_72h	CTRL_3h	16.6
24.29_743.2418 *m*/*z*	743.2418	1	24.29	0.61	0.704	0.838	1.38	AZO_3h	AZO_96h	11.4
3.71_380.1017 *m*/*z*	380.1017	2	3.71	0.32	0.978	>0.9995	4.22	CTRL_96h	CTRL_24h	18.3

Z = charge; RT = retention time; PW = peak width; FC = Fold change; HM = highest mean; LM = lowest mean.

**Table 2 molecules-24-02473-t002:** Mass spectrometric parameters used for the Multiple Reaction Monitoring-Information Dependent Acquisition (MRM-IDA) spectra acquisition of glutathione conjugate of azoxystrobin.

Scan type	Q1	Q3	DP (V)	EP (V)	CE (V)	CXP (V)	CUR (psi)	CAD	IS (V)	TEM (°C)	GS 1 (psi)	GS 2 (psi)
MRM	711	404	60	10	30	13	35	High	4300	450	90	50
	711	372	60	10	30	13	35	High	4300	450	90	50
	711	582	60	10	30	13	35	High	4300	450	90	50

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
