# Peer review of "Identification of Azoxystrobin Glutathione Conjugate Metabolites in Maize Roots by LC-MS"

_molecules, 2019, doi:10.3390/molecules24132473_

Round 1
Reviewer 1 Report
The manuscript entitled “Identification of azoxystrobin glutathione conjugate metabolites in maize roots by LC-MS” identified the possible metabolites of azoxystrobin in maize roots using LC-MS methods. It could be considered for publication in Molecules after addressing the following questions.
1) Line 24: what was the meaning of “MSe”?
2) Figure 2: was the unit of retention time seconds or minutes?
3) Figure 3: what was the top figure?
4) Figure 4A: how did you obtain this figure? At which retention time?
5) Figure 4B and 5A: were these two figures the MS spectrum of the potential azoxystrobin at RT 18.36 min? Why no m/z 654 was observed in Figure 5A?
6) Figure 5B: the evidence for the identification of potential azoxystrobin conjugate at RT 18.36 min was not enough, because only the ion at m/z 372 was explained. Why the ions at m/z 344 and 329 were not observed (Figure 2, the main fragmentation ions of azoxystrobin)? Moreover, the ions produced from the neutral loss of glycine on GSH were not detected.
7) Line 225-226: why was the retention time of demethylated azoxystrobin-GSH conjugate (30.55) longer than that of azoxystrobin-GSH conjugate (18.36) on C18 column?
8) Line 291-293: as you mentioned “a glutathione (GSH) conjugate of azoxystrobin has been identified in rats where the sulphur is conjugated to the phenolic ring in the ortho position relative to the cyanonitrile group”, why did you consider that GSH connected with CN group? I didn’t find any MS/MS evidence, which could indicate this connection.
Author Response
We are grateful to the reviewer 1 for bringing to our attentions the points that we have now tried to fix.
Question 1) Line 24: what was the meaning of “MSe”?
Reply 1) Added in Abbreviations: MSe, Waters all ion fragmentation DIA acquisition method. It acquires data in three channels: MS1 or survey at low CE, i.e. from 1-5 eV; MS2: ramp or fixed CE fragmentation MS/MS; MS3 channel is MS/MS of a lock mass standard peptide compound needed for internal calibration every 1.5-3 minutes. The MSe data format is raw and Waters software convert it to ppm error corrected centroid data for further analysis.
Question 2) Figure 2: was the unit of retention time seconds or minutes?
Reply2) That is right, thanks, it was a mistake writing seconds. Of course, it is related to minutes. The legend of Figure 2 has been corrected.
Question 3) Figure 3: what was the top figure?
Reply 3) It was forgotten to add which trace belong to which. The follwing info has been added to the legend of Figure 3: A comparative extracted chromatogram for control untreated (top MS/MS trace) and treated with azoxystrobin (lower MS/MS trace) extracted and analyzed after 96h.
Question 4) Figure 4A: how did you obtain this figure? At which retention time?
Reply 4) his figure is related to the “possible identification of potential azoxystrobin conjugate at RT 18.10-18.36 minutes” as it is stated in the legend of Figure 4. The 4A is MS1 (survey at the same RT) of the m/z region of interest for the control untreated as also stated in the legend. This trace is a grouped function trace obtained by MassLynx from RT18.10 up to RT 18.36.
Question 5) Figure 4B and 5A: were these two figures the MS spectrum of the potential azoxystrobin at RT 18.36 min? Why no m/z 654 was observed in Figure 5A?
Reply5) In Figure 4A/B the traces are pooled from traces obtained by MassLynx from RT18.10 up to RT 18.36.
Meanwhile the Trace of Figure 5A comes from the RT18.36 alone. In this framed RT the specie m/z 654 is not present, probably because it has been eluted before at RT 18.10 which could indicate that the fragmentation of the glycil moiety could be happening in vivo by the action of a carboxypeptidase (Wolf et al., 1996) or a phytochelatin synthase with carboxypeptidase activity (Beck et al., 2003) and not during the experimental condition of extraction and analysis. A Text related to this has been added to the discussions.
Beck A, Ledzian L., Oven M., Christmann A and Grill E., (2003) Phytochelatin
synthase catalyses key step in the turnover of glutathione conjugates.
Phytochemistry 62: 423-431.
Wolf A.E., Dietz K.J. and Schroder P. (1996) Degradation of glutathione S-conjugates
by a carboxypeptidase in the plant vacuole. FEBS letters 384: 31-34.
Question 6) Figure 5B: the evidence for the identification of potential azoxystrobin conjugate at RT 18.36 min was not enough, because only the ion at m/z 372 was explained. Why the ions at m/z 344 and 329 were not observed (Figure 2, the main fragmentation ions of azoxystrobin)? Moreover, the ions produced from the neutral loss of glycine on GSH were not detected.
Reply6) The compound at RT 18.36 min was identified based on possible attachment structure drawn by chemical software and extrapolation of their possible m/z values.
The compound with m/z 711 could have different fragmentations profiles that azoxystrobin itself since the GSH or part of it can remain attached to the CN group. This explain why the structure with m/z 344 and 329 could not be seen. Anyhow its chemical synthesis and MS/MS fragmentation could solve this issue.
Here are the detailed chemical names of the compound with m/z 711:
InChI String InChI=1S/C32H34N6O11S/c1-46-15-20(32(45)47-2)18-7-3-5-9-23(18)48-26-13-27(37-17-36-26)49-24-10-6-4-8-19(24)29(34)50-16-22(30(42)35-14-28(40)41)38-25(39)12-11-21(33)31(43)44/h3-10,13,15,17,21-22,34H,11-12,14,16,33H2,1-2H3,(H,35,42)(H,38,39)(H,40,41)(H,43,44)/b20-15+,34-29?
InChI Key IGJUSXVHXDCBIH-BFMANGOVSA-N
ExactMass 710.200626658
IUPAC Name 2-amino-5-[[2-(carboxymethylamino)-1-[[2-[6-[2-[(E)-2-methoxy-1-methoxycarbonyl-vinyl]phenoxy]pyrimidin-4-yl]oxybenzenecarboximidoyl]sulfanylmethyl]-2-oxo-ethyl]amino]-5-oxo-pentanoic acid
SMILES CO\C=C(\C(=O)OC)/c1ccccc1Oc2cc(Oc3ccccc3C(=N)SCC(NC(=O)CCC(N)C(=O)O)C(=O)NCC(=O)O)ncn2
C32 H34 N6 O11 S
Mol weight: 710.71096
Question 7) Line 225-226: why was the retention time of demethylated azoxystrobin-GSH conjugate (30.55) longer than that of azoxystrobin-GSH conjugate (18.36) on C18 column?
Reply 7) The trap column and the capillary column used here are mixed bed silica and C18, RP-C18 HHS (High Strength Silica T3 chemistry, Waters) and hence the elution profile does not follow the logic of C18 RP columns. It is used for both polar and apolar compound detection.
Question 8) Line 291-293: as you mentioned “a glutathione (GSH) conjugate of azoxystrobin has been identified in rats where the sulphur is conjugated to the phenolic ring in the ortho position relative to the cyanonitrile group”, why did you consider that GSH connected with CN group? I didn’t find any MS/MS evidence, which could indicate this connection.
Reply8) We have made the extra pdf file with 5 figure explaining the possible route of the attachement with the -CN group:
As first (see figure 1 and two of the extra pdf file), we did screen but could not find for the rat type of GSH conjugate at the phenolic ring with m/z 708.18 (M+H+=709.18)
InChI String InChI=1S/C32H32N6O11S/c1-46-15-20(32(45)47-2)18-6-3-4-7-23(18)48-27-12-28(37-17-36-27)49-24-8-5-9-25(19(24)13-33)50-16-22(30(42)35-14-29(40)41)38-26(39)11-10-21(34)31(43)44/h3-9,12,15,17,21-22H,10-11,14,16,34H2,1-2H3,(H,35,42)(H,38,39)(H,40,41)(H,43,44)/b20-15+/t21-,22-/m0/s1
and with 694.16 (M+H+=695.16) for the demethylated form
InChI String InChI=1S/C31H30N6O11S/c1-46-31(45)19(14-38)17-5-2-3-6-22(17)47-26-11-27(36-16-35-26)48-23-7-4-8-24(18(23)12-32)49-15-21(29(42)34-13-28(40)41)37-25(39)10-9-20(33)30(43)44/h2-8,11,14,16,20-21,38H,9-10,13,15,33H2,1H3,(H,34,42)(H,37,39)(H,40,41)(H,43,44)/b19-14+/t20-,21-/m0/s1
Hence, after we could not find any peak related to their M+H+ species, the only differentially peaks differentially present were, of course the demethylated ones, but beside it, the four ones of which the m/z 711 ones was the most abundant in the azoxystrobin treatment versus the untreated compound. Since we do not have any standard for it the identification based on the MS1 and MS/MS with few fragments matching the ones from azoxystrobin could be questionable as you said. Anyway, the GSH typical neutral loss of 129 (Aspartyl residue), the glycil loss by enzymatic activity in vivo, are the only evidence that the compound with m/z 711 could correspond to the GSH attachment to the -CN group, which passed from cyano to acryl-nitrile without loss with the proton being passed to the nitrogen (Z group). Beside the proposed structure matches the chemical structure as M+H+ and observed neutral losses.
Secondly, we have produced in the text and in the discussion explained with references the possible metabolic presence of the GSH-Azoxystrobin conjugate whereas the glycine loss is actually an in vivo degradation for storage purposes, as shown in the extra figures present in the pdf file (figure 3 and 4). The figure 5 group all the possible modification that a GSH Xenobiotic conjugate can undergone in the plant system with the related enzymes and losses depicted.
Further experiments as the isolation of the GSH transferase and in vitro coupling could give an example of the 711 m/z compound as well as its chemical synthesis and MS/MS fragmentation.

Reviewer 2 Report
The paper faces the Xenobiotic detoxification in plant for azoxystrobin a fungicide accumulated over time studying the metabolita conjugates due to GSH in maize roots .
The main detoxified compound was the methyl ester hydrolysis product
thought to be inactive followed by the glutathione conjugated.
The glycosylated form of azoxystrobin was also in a minor amount.
The identification of these analytes were done by differential untargeted
metabolomics analysis by spectral quantification and MS/MS confirmation and by high resolution LC-MS methods. The experiments confirmed the
structures of these new azoxystrobin GSH conjugates.
My opinion is tha the the paper is s well designesd, well executed and , then , deserves publication. Only few small correction are suggested to less familial reader.
A) a chemical structure is to be inserted as Figure 1 at the beginning.
B) Secondly in the introduction a short description how the GSH enzymes perform their detoxification action ( a short description of the molecular mechanism).
C) At rows 64-66 some more words about the results reported in ref 6 should be added.
D) Fig. 6 it is not particularly clear and the legend(s) do not help to understand easily the content.
E) Some typing errors are along the text ( i.e. glutathionse).
Author Response
We are also grateful to the reviewer 2 for letting us noticing errors and typos to fix.
Question A) a chemical structure is to be inserted as Figure 1 at the beginning.
Reply A) thanks for the suggestion and it has been inserted in Figure 1 with more info in the label for what is concerned its chemical IUPAC name and exact mass.
Question B) Secondly, in the introduction a short description how the GSH enzymes perform their detoxification action ( a short description of the molecular mechanism).
Reply B) Thanks, we have now implemented in the introduction the description of the Glutathione Transferase enzymes, with a short description of the mechanism. Since the argument is vast and the classes of the enzymes are also a lot, we have not extended further, since also it is under current investigation from our site the chasing for hunting which GST enzyme is performing the Azoxystrobin detoxification. The leaving group Z is the Nitrogen that cannot leave because in triple bond, but it reduce the number of bond and acquire the proton from the GSH.
Question C) At rows 64-66 some more words about the results reported in ref 6 should be added.
Reply C) Thanks, we have briefly mentioned that our previous results are merely to establish a system for in vitro study the fungicide feeding to the roots, extraction and quantification, but here we have also been able to detect minor GSH conjugated that we were not able to detect before, as the main metabolite remain to be the demethylated form (azoxystrobin free acid).
Question D) Fig. 6 it is not particularly clear and the legend(s) do not help to understand easily the content.
Reply D) Thanks, more info has been added. To the text related to the 2D or 3D visualization performance of the software offering a visual increasing trend of the peak isotopes of the m/s species in question.
Question E) Some typing errors are along the text ( i.e. glutathionse).
Reply E) Thanks, Typing errors have been corrected.
Round 2
Reviewer 1 Report
I have no further comment.